# Quantification of Lung Perfusion Blood Volume in Dual-Energy Computed Tomography in Patients with Pulmonary Hypertension

**DOI:** 10.3390/life12050684

**Published:** 2022-05-05

**Authors:** Satoko Ugawa, Satoshi Akagi, Kentaro Ejiri, Kazufumi Nakamura, Hiroshi Ito

**Affiliations:** 1Department of Cardiovascular Medicine, Okayama University Graduate School of Medicine, Dentistry, and Pharmaceutical Sciences, Okayama 700-8558, Japan; satokougauga@hotmail.com (S.U.); eziken82@gmail.com (K.E.); ichibun@cc.okayama-u.ac.jp (K.N.); itomd@md.okayama-u.ac.jp (H.I.); 2Department of Cardiology, Kagawa Prefectural Central Hospital, Takamatsu 760-8557, Japan

**Keywords:** pulmonary vascular bed, pulmonary arterial hypertension, lung perfusion scintigraphy

## Abstract

Dual-energy computed tomography (DECT) is a promising technique for the assessment of the lung perfused blood volume (LPBV) in the lung parenchyma. This study was performed to compare the LPBV by DECT of patients with pulmonary hypertension (PH) and controls and to evaluate the association between the LPBV and the perfusion ratio derived by lung perfusion scintigraphy. This study involved 45 patients who underwent DECT (25 patients with PH and 20 controls). We measured the total LPBV and distribution of the LPBV in each lung. The total LPBV was significantly lower in the PH group than the control group (38 ± 9 vs. 45 ± 8 HU, *p* = 0.024). Significant differences were observed between the LPBV of the upper lung of the PH and control groups (34 ± 10 vs. 47 ± 10, *p* = 0.021 and 37 ± 10 vs. 47 ± 8, *p* < 0.001). A significant correlation was observed between the LPBV and the lung perfusion scintigraphy. A lower total LPBV and lower LPBV of the upper lung as detected by DECT might be specific findings of PH.

## 1. Introduction

Pulmonary hypertension (PH) is caused by pulmonary vascular dysfunction due to stenosis and obstruction secondary to vasoconstriction, vascular remodeling, and thrombi, which leads to an increased pulmonary arterial load and right ventricular dysfunction. In the current guidelines, PH is defined as an elevation of the mean pulmonary artery pressure (PAP) (>25 mmHg) and pulmonary vascular resistance as measured by right heart catheterization [1]. However, previous studies suggested that the recognition of pathophysiological pulmonary vascular dysfunction is prior to an elevation of PAP. A reduction of the pulmonary vascular bed is, therefore, one of the initial findings of PH [2]. In other words, assessment of the pulmonary vascular bed may be useful for early diagnosis of PH and estimation of disease progression.

Lung perfusion scintigraphy is a clinical tool used to assess pulmonary blood flow; however, it does not allow for quantitative evaluation. Multidetector computed tomography enables anatomical evaluation based on the central pulmonary artery size and right ventricular morphology, but it does not provide information on perfusion [3]. Previous studies showed that dual-energy computed tomography (DECT), which is an imaging technique involving two types of X-ray with different tube voltages, is promising for the assessment of the lung perfused blood volume (LPBV) by measurement of the distribution of iodine contrast medium in the lung parenchyma [4,5,6,7]. We hypothesized that measurement of the LPBV by DECT is useful for assessment of the pulmonary vascular bed in patients with PH. The aims of this study were to compare the LPBV as measured by DECT of patients with PH and controls and to evaluate the association between the LPBV as measured by DECT and lung perfusion scintigraphy. 

## 2. Materials and Methods

### 2.1. Study Design

This was a retrospective, observational study. The study conformed to the principles outlined in the Declaration of Helsinki. It was approved by the institutional review board and ethics committee of Okayama University (no. 1912-017). The requirement for informed consent was waived because of the low-risk nature of this retrospective study and the inability to obtain consent directly from all subjects. Instead, we extensively announced this study protocol at Okayama University Hospital and on our website (http://www.hsc.okayama-u.ac.jp/ethics/koukai/jyunkan/index.html (accessed on 31 March 2022)) and gave patients the opportunity to withdraw from the study.

### 2.2. Patient Population

In total, 25 patients with PH who underwent DECT using an iodine contrast agent and right heart catheterization at Okayama University Hospital from August 2013 to August 2014 were included in this study. PH was defined as a mean PAP of ≥25 mmHg on right heart catheterization. Pulmonary arterial hypertension (PAH) was defined as a mean PAP of ≥25 mmHg, pulmonary artery wedge pressure of ≤15 mmHg, and pulmonary vascular resistance of ≥3 Wood units on right heart catheterization [1]. Pulmonary veno-occlusive disease was defined as CT findings of ground glass opacity, severe hypoxia, and low diffusing capacity for carbon monoxide. The control group comprised 20 patients with no history of PH and no suspected venous thrombosis or pulmonary thromboembolism. Blood tests and echocardiography were performed within 1 month of DECT in both groups. Right heart catheterization was performed within 1 month of DECT to evaluate hemodynamics in the PH group. Information about pretreatment with PAH-specific drugs was obtained from the patients’ medical records.

### 2.3. Data Collection and Measurement

The LPBV was estimated using the distribution of contrast medium in the pulmonary artery on CT scans. We compared the total LPBV and the distribution of LPBV in each lung of the PH group and control group. The patients underwent a CT scan in the supine position, and the LPBV was evaluated. LPBV measurement was performed using the dual-energy mode of a 2- × 128-slice dual-source CT scanner (SOMATOM Definition Flash; Siemens Healthineers, Erlangen, Germany). The first detector had a 50 cm field of view, and the second detector had a 26 cm field of view. The two tube voltages were set to 80 and 140 kVp, the detector collimation was 32 × 0.6 mm, the gantry rotation speed was 0.33 s per rotation, and the pitch was 0.5. After securing the route to the right median cubital with a 22-gauge indwelling needle, 100 mL of the low-osmotic, nonionic iodine contrast agent iopamidol (Iopamiron 370) was administered, followed by injection of 30 mL of physiological saline at 3 mL/s. At 25 s after injection of the contrast medium, images were taken from the diaphragm level to the apex level in the caudal direction. The CT dose index was 10.38 mGy/cm, and radiation exposure was comparable with that of a routine CT angiography. Images were reconstructed with a 1 mm slice thickness and spacing and created using a specific medium-soft convolution kernel D40 that did not affect the shape of other materials. The spatial resolution of DECT was a 512 × 512 matrix. When lung perfusion CT was performed, 80 and 140 kVp tube voltages were used, and lung perfusion CT images were obtained semi-automatically with the attached workstation (syngo.CT Dual Energy software (syngo 2004A (VD10B)); Siemens Healthineers, Erlangen, Germany). The obtained lung perfusion CT images were analyzed using Ziostation2 (Ziosoft Inc., Newark, CA, USA). The LPBV was automatically calculated for the whole lung, the left and right lungs, and the upper, lower, middle, and lower lungs (Figure 1). We defined LPBV ratio as the value of LPBV divided by total LPBV to avoid the effects of cardiac output. Furthermore, the correlations of the CT-estimated LPBV with hemodynamic parameters measured by right heart catheterization were investigated in this study. 

Lung perfusion scintigraphy was performed in 20 patients in the PH group to evaluate the correlation between lung perfusion scintigraphy and the LPBV from DECT. 99mTc-labeled macroaggregated albumin was injected into an antecubital vein with the patient in the supine position. Immediately after tracer administration, lung scanning was performed using a gamma camera system equipped with a high-resolution collimator. The perfusion ratio was automatically obtained by the workstation. Right heart catheterization was performed to access hemodynamic parameters in the PH group. Cardiac output and pulmonary vascular resistance were calculated by Fick’s method [8]. Transthoracic echocardiography was performed on all patients in the PH and control groups. Measurement of the heart chambers and Doppler imaging were performed according to the American Society of Echocardiography guidelines [9,10]. 

### 2.4. Statistical Analysis

All data are expressed as mean with standard deviation or median (interquartile range). Continuous variables were compared between the groups with the Mann–Whitney U test or Kruskal–Wallis test for non-normally distributed data and with an unpaired *t*-test or analysis of variance for normally distributed data. Categorical variables were compared with the chi-square test. The association between the LPBV and data from right heart catheterization was evaluated using Spearman’s rank correlation coefficient. Statistical significance was defined as a two-tailed *p* value of <0.05. All statistical analyses were performed with SPSS software version 25.0 (IBM Corp., Armonk, NY, USA) and R version 3.6.3 (The R Foundation for Statistical Computing, http://www.R-project.org (accessed on 31 March 2022)).

## 3. Results

### 3.1. Patient Characteristics in PH and Control Groups

The baseline characteristics of the patients in the PH group and control group are shown in Table 1. The 25 patients in the PH group comprised 24 patients with PAH and one patient with suspected pulmonary veno-occlusive disease. There were no significant differences in age, sex, body mass index, heart rate, blood pressure, oxygen saturation, hemoglobin concentration, renal function, or liver function between the PH and control groups. However, the serum albumin concentration was significantly different between the two groups (4.1 ± 0.5 vs. 3.7 ± 0.58 g/dL, *p* = 0.03). Among the echocardiographic parameters, the left ventricular ejection fraction and tricuspid regurgitation pressure gradient were significantly higher in the PH group than in the control group (68% ± 7% vs. 60% ± 6%, *p* = 0.043 and 58 ± 22 vs. 17 ± 3 mmHg, *p* < 0.001). No significant difference in the stroke volume was observed between the two groups. 

In the PH group, idiopathic PAH, PAH associated with connective tissue disease, and PAH associated with congenital heart disease accounted for most of the cases. The proportion of patients with World Health Organization functional class III and IV was 32%. The mean pulmonary artery wedge pressure was 9 ± 4 mmHg, mean PAP was 39 ± 10 mmHg, and mean pulmonary vascular resistance was 7.4 ± 4.5 Wood units.

### 3.2. Measurement of LPBV by DECT

Representative images of the LPBV are shown in Figure 1. 

The LPBV by DECT in the total, left, and right lung was significantly lower in the PH group than in the control group (Table 2 and Figure 2). 

The LPBVs in the bilateral upper lungs were significantly lower in the PH group than in the control group (right lung, 37 ± 10 vs. 47 ± 8; *p* < 0.001 and left lung, 34 ± 10 vs. 47 ± 10; *p* < 0.001), while those in the middle and lower lungs of the bilateral lungs were not significantly different between the two groups (Figure 3). 

No significant difference was observed in the LPBV ratio of both sides of the lung between the PH and control groups (Figure 4). 

In both sides of the lung, however, the LPBV ratio of the upper lung was significantly lower in the PH group than in the control group (right lung, 0.95 ± 0.07 vs. 1.05 ± 0.10; *p* < 0.001 and left lung, 0.89 ± 0.14 vs. 1.05 ± 0.08; *p* < 0.001) (Table 2 and Figure 5). Conversely, the LPBV ratio of the middle or lower lung was significantly higher in the PH group than in the control group.

### 3.3. Association between LPBV Measured by DECT and Pulmonary Blood Flow Scintigraphy

Twenty patients in the PH group underwent both DECT and pulmonary blood flow scintigraphy. Fifteen patients underwent DECT within one week of lung perfusion scintigraphy. Three patients underwent DECT within one month of lung perfusion scintigraphy. Two patients underwent DECT after one year of lung perfusion scintigraphy. Figure 6 shows the association of the LPBV ratio on both sides to the total lung with the perfusion ratio derived by pulmonary blood flow scintigraphy. There were significant correlations between the LPBV ratios of the right and left sides to the total lung and the perfusion ratio derived by pulmonary blood flow scintigraphy (correlation coefficients, 0.59 and 0.69; *p* < 0.001 and *p* < 0.001). 

## 4. Discussion

In the present study, the LPBV by DECT was significantly lower in the PH group than in the control group. We found a significant difference in the distribution of contrast medium in the upper lung on the DECT scan between the two groups. Furthermore, the LPBV ratio of the upper lung was significantly lower and the LPBV ratio of the middle or lower lung was significantly higher in the PH group than in the control group. The LPBV ratios measured by DECT were well correlated with lung perfusion flow scintigraphy in patients with PH. This is the first study to evaluate the association of the LPBV measured by DECT with specific findings for patients with and without PH. 

PH is caused by pulmonary vasculature abnormalities such as vascular endothelial damage, cell proliferation, vasoconstriction, and occlusion in the peripheral pulmonary vasculature [11]. The Heath–Edwards (HE) classification is often used to assess the progression of pulmonary vascular lesions in patients with PAH [12]. A rat model similar to human PAH showed that the remodeling of HE-class-I to -III lesions occurs over time from the early stage to the middle stage of the disease, and the HE-class-IV plexiform lesion occurs only in the late stage [13]. HE-class-I and -II lesions are characterized by isolated media hypertrophy of the pulmonary artery. This remodeling results from vasospasm or dilation of the pulmonary arteries in early-stage PH. In patients with PAH, remodeling of the pulmonary arteries primarily leads to reduced blood flow in the pulmonary circulation. Because the progression of pulmonary artery remodeling is heterogenous in each lobe in patients with PAH, a mottled pattern is observed on lung perfusion scintigraphy [14,15,16]. In the present study, the LPBV by DECT was significantly lower in patients with than without PH. This finding suggests lower blood flow in the pulmonary circulation in patients with PH, as shown by lung perfusion scintigraphy [6,7,16]. The LPBVs in the upper lungs were significantly lower in the PH group than in the control group. Furthermore, the LPBV ratios of the upper lung were significantly lower in the PH group than in the control group. Conversely, the LPBV ratios of the middle or lower lung were significantly higher in the PH group than in the control group. These results suggest that a decrease in the perfused blood flow in the upper lung could lead to a relative increase in blood flow in the middle and lower lungs compared with the total lung. We consider that lower blood flow in the upper lung may be a specific feature that reflects pulmonary vascular remodeling in patients with PH. In a previous study performed to assess the gravity-dependent redistribution of pulmonary perfusion, Lau et al. reported that patients with precapillary PH displayed pronounced attenuation in the normal, gravity-dependent redistribution of lung perfusion compared with healthy controls, and this attenuation was significantly associated with prognostic parameters such as the 6 min walk distance, functional class, and tricuspid annular plane systolic excursion [17]. The authors suggested that this finding reflected lower pulmonary vascular reserve in patients with rather than without PH. In the present study, the lower LPBV and the LPBV ratio of the upper lung in the DECT scan may also reflect pulmonary vascular remodeling, and measurement of these indices might, therefore, be a promising noninvasive diagnostic tool for PH. However, further investigation using a larger sample size is needed. 

In the present study, there was a significant correlation between the perfusion ratio derived by pulmonary blood flow scintigraphy and the LPBV ratio by DECT. This result suggests that the LPBV by DECT could be an alternative imaging modality with which to diagnose PH instead of pulmonary blood flow scintigraphy. The spatial resolution of DECT is superior to that of lung perfusion scintigraphy; therefore, the LPBV might allow for more accurate evaluation of pulmonary perfusion than scintigraphy. Of course, the perfusion ratio of pulmonary blood flow scintigraphy differs from the LPBV derived from DECT. Pulmonary perfusion scintigraphy is an imaging modality that allows for visualization of the pulmonary blood flow distribution and is the gold standard for differential diagnosis of PH [18]. The finding of chronic thromboembolic PH on pulmonary blood/ventilation scintigraphy indicates the presence of a regional blood flow distribution defect with no abnormal ventilation distribution. In the diagnosis of chronic thromboembolic PH, iodine distribution maps on DECT enable segmental blood flow evaluation equivalent to that by pulmonary blood flow scintigraphy [19,20]. In previous reports on pulmonary blood flow scintigraphy and DECT, DECT was found to be more useful than pulmonary blood flow scintigraphy for occlusion of relatively large blood vessels compared with the main lesion of PAH [21]. Pulmonary blood flow scintigraphy plays an important role in the diagnosis of PH; however, it is often difficult to perform pulmonary blood flow scintigraphy in patients with unstable conditions, such as patients whose condition may be adversely affected by breath holding or exacerbated by decreased pulmonary blood flow due to macroaggregated albumin [22,23]. DECT has a shorter inspection time and might be more suitable for patients with PAH who can tolerate the administration of contrast media. We believe that obtaining the LPBV by DECT is a promising technique with which to estimate pulmonary blood flow in undiagnosed patients in the acute phase. 

This study had several limitations. This was a retrospective, single-center study, and the patient population was small. Several factors might have affected the LPBV. First, several etiologies of PAH were included in the study. Second, about 60% patients received PAH-specific drugs. Third, severe PH impairs lungs. Fourth, the PBV of the upper lung is affected by artifacts of the iodine contrast agent in the superior vena cava and the subclavian vein [24,25], although we used the right median cubital to prevent the arrival delay of contrast medium and to reduce artifacts in the superior vena cava and the subclavian vein. There were time differences between DECT and pulmonary blood flow scintigraphy, which would have biased the results. To confirm the present results, a large-scale cohort study using a larger registry is required. However, LPBV obtained from DECT is an excellent diagnostic imaging technique that could contribute to the diagnosis of PH because it can visualize and quantify regional perfusion decline in addition to anatomical information obtained from good spatial resolution. Especially in situations where pulmonary blood flow scintigraphy cannot be performed, it might be an alternative imaging examination for evaluating regional perfusion. Although it is a small study, we hope the present study will be useful in clinical practice.

## 5. Conclusions

In conclusion, the pulmonary vascular bed can be noninvasively evaluated by quantitative assessment of pulmonary perfusion using DECT. Measurement of the LPBV by DECT is a promising imaging modality with which to assess the pulmonary vascular bed in patients with PH. Further investigation with a larger sample size is warranted.

## Figures and Tables

**Figure 1 life-12-00684-f001:**
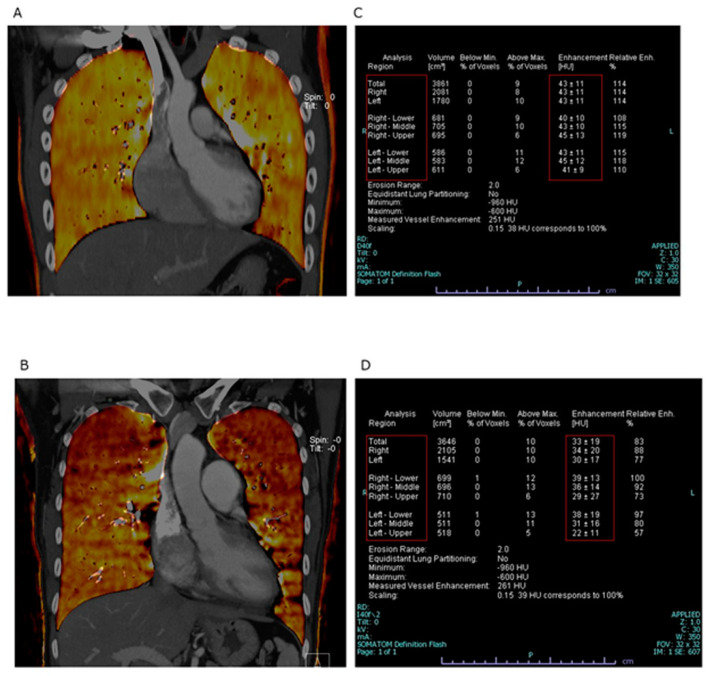
Representative images of LPBV. (**A**,**B**) Coronal images of dual-energy computed tomography iodine map. (**A**) Control. (**B**) PH. (**C**,**D**) The value of the lung perfused blood volume was automatically calculated by the workstation. (**C**) Control. (**D**) PH.

**Figure 2 life-12-00684-f002:**
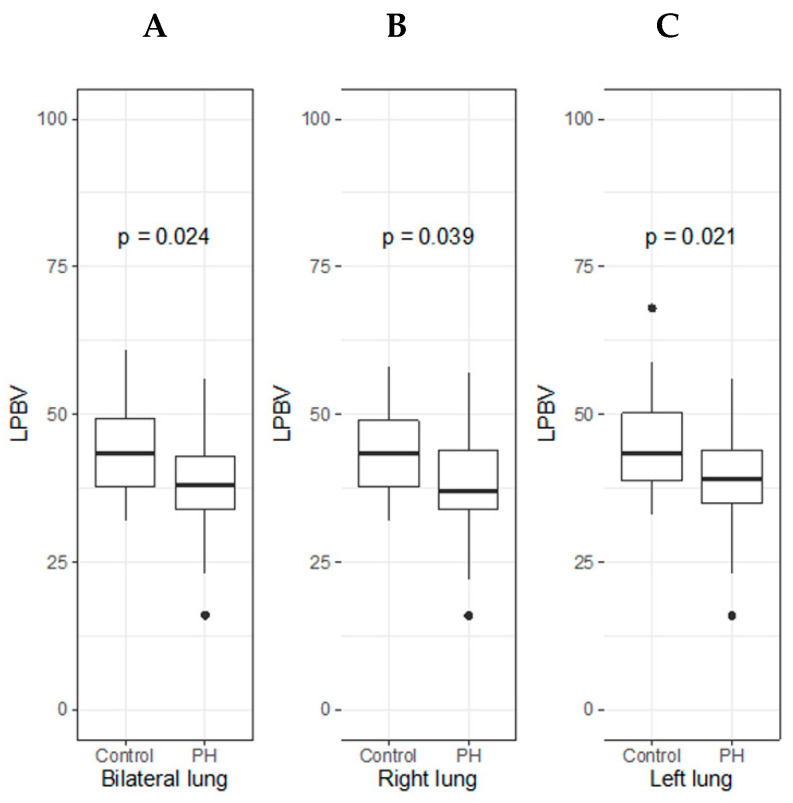
LPBV of total, right, and left lung between PH group and control group. Box-and-whisker plots display the median, interquartile range, minimum, and maximum values of the LPBV in each side of the lung ((**A**) bilateral; (**B**) right; and (**C**) left) between the PH group and control group. In each side of the lung, the LPBV was significantly lower in the PH group than control group.

**Figure 3 life-12-00684-f003:**
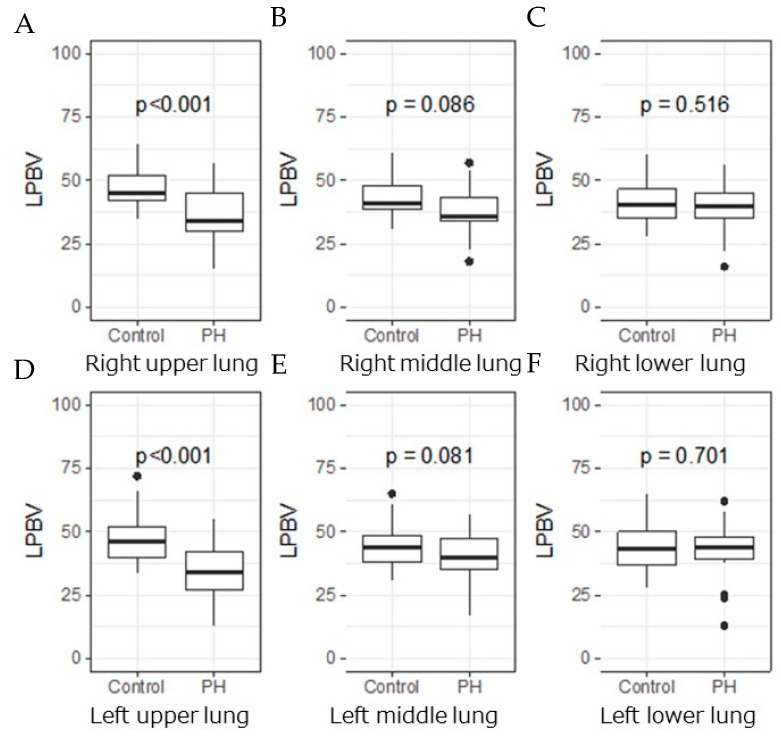
LPBV in each lung of PH group and control group. Box-and-whisker plots display the median, interquartile range, minimum, and maximum values of the LPBV in each lung ((**A**) right upper; (**B**) right middle; (**C**) right lower; (**D**) left upper; (**E**) left middle; and (**F**) left lower) of the PH group and control group. A significant difference was observed between the LPBV of the upper lung of the PH group and control group (right lung, 37 ± 10 vs. 47 ± 8; *p* < 0.001 and left lung, 34 ± 10 vs. 47 ± 10; *p* < 0.001), while no difference was observed in the middle and lower lobes.

**Figure 4 life-12-00684-f004:**
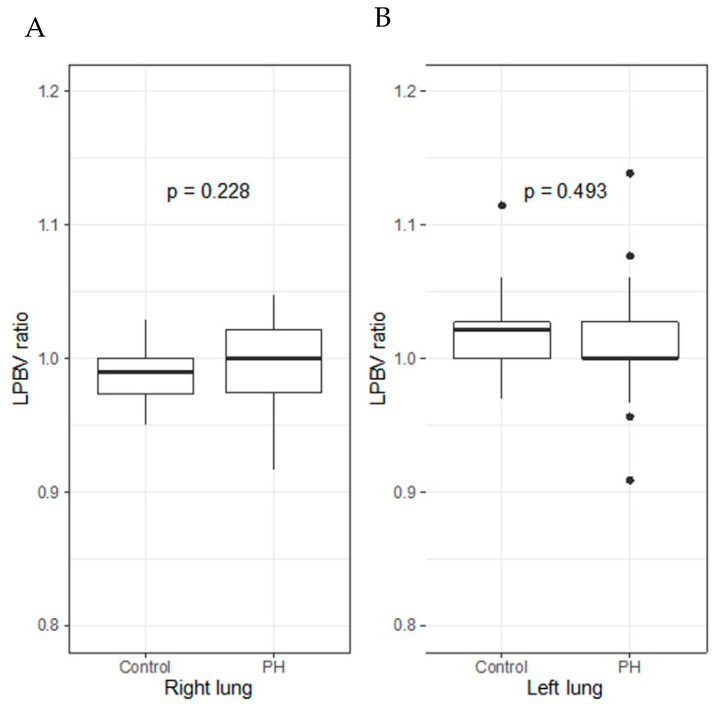
LPBV ratios of both sides of lung in PH group and control group. Box-and-whisker plots display the median, interquartile range, minimum, and maximum values of the LPBV ratios of each side of the lung ((**A**) right; (**B**) left) for the PH group and control group. There were no significant differences between the LPBV ratio of the right and left lung of the PH group and control group.

**Figure 5 life-12-00684-f005:**
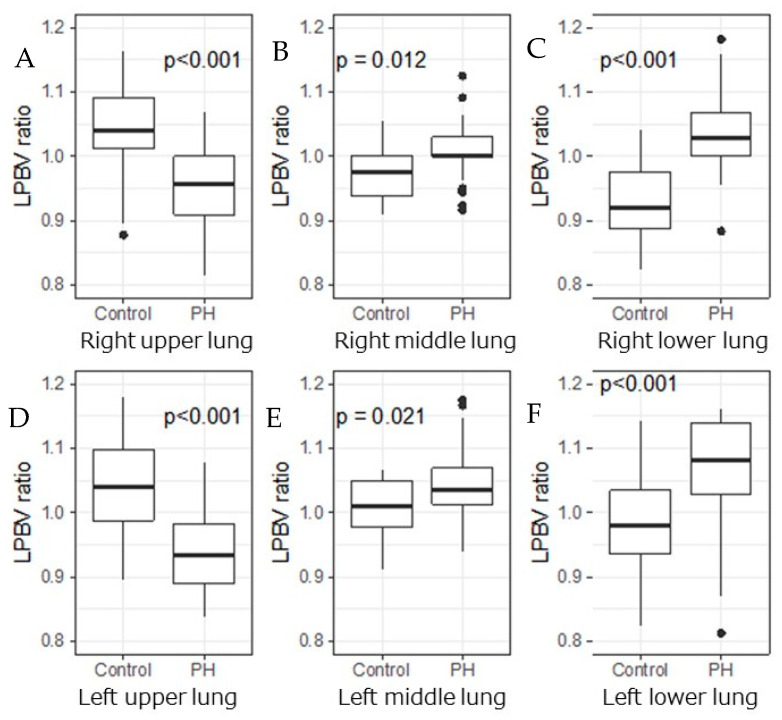
LPBV ratios of each lung in PH group and control group. Box-and-whisker plots display the median, interquartile range, minimum, and maximum values of the LPBV ratios of each lung ((**A**) right upper; (**B**) right middle; (**C**) right lower; (**D**) left upper; (**E**) left middle; and (**F**) left lower) for the PH group and control group. In both sides of the lung, the LPBV ratios of upper lung were significantly lower in the PH group than in the control group (right lung, 0.95 ± 0.07 vs. 1.05 ± 0.10; *p* < 0.001 and left lung, 0.89 ± 0.14 vs. 1.05 ± 0.08; *p* < 0.001) (see also Table 2). Conversely, the LPBV ratios of the middle or lower lung were significantly higher in the PH group than in the control group.

**Figure 6 life-12-00684-f006:**
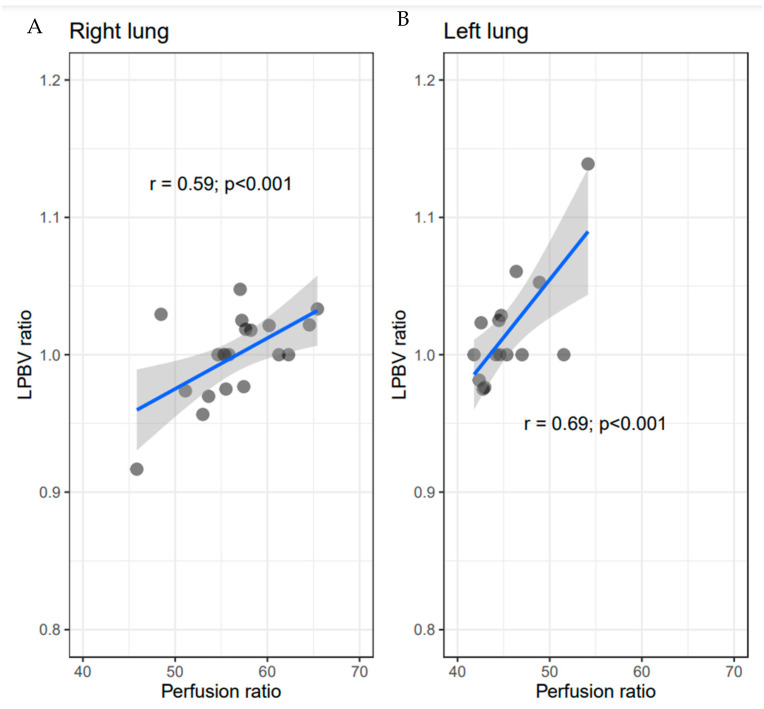
Correlations between LPBV ratios in DECT and perfusion ratios in lung perfusion scintigraphy. Scatter plots display associations between the LPBV ratio on both sides to the total LPBV with the perfusion ratio of pulmonary blood flow scintigraphy in 20 patients of the PH group ((**A**) right; (**B**) left). Blue lines and gray areas display linear regression lines and their 95% confidence intervals. There were significant correlations between the LPBV ratios and perfusion ratios in both the right lung and left lung (r = 0.59, *p* < 0.001 and r = 0.69, *p* < 0.001, respectively).

**Table 1 life-12-00684-t001:** Characteristics of patients in PH group and control group.

	PH Group N = 25	Control Group N = 20	*p* Value
Age (years)	44 ± 16	42 ± 13	0.622
Female (%)	17 (68)	17 (34)	0.183
BMI (kg/m^2^)	21 ± 3	22 ± 4	0.569
Heart rate (bpm)	76 ± 13	76 ± 15	0.886
Blood pressure (mmHg)			
Systolic	113 ± 17	114 ± 10	0.843
Diastolic	68 ± 14	70 ± 7	0.537
SpO_2_ (%)	97 ± 3	98 ± 1	0.114
Blood examination			
BNP (pg/mL)	196 ± 346	14 ± 9	0.015
Echocardiography			
TRPG (mmHg)	59 ± 20	17 ± 3	<0.001
TAPSE (mm)	19 ± 5		
Etiology, n (%)		NA	NA
Idiopathic	7 (28)		
Collagen disease	9 (36)		
Congenital heart disease	6 (24)		
Others	3 (12)		
WHO functional class		NA	NA
II	17 (68)		
III	6 (24)		
IV	2 (8)		
Pulmonary function test		NA	NA
FEV1 (%)	78.8 ± 9.6		
FVC (%)	85.5 ± 13.4		
%DLCO	54 ± 18		
Hemodynamics		NA	NA
SPAP (mmHg)	59 ± 16		
DPAP (mmHg)	26 ± 7		
mPAP (mmHg)	39 ± 10		
PAWP (mmHg)	9 ± 4		
RAP (mmHg)	5 ± 3		
CO (L/min)	5.1 ± 2.3		
CI (L/min/m^2^)	3.2 ± 1.4		
PVR (wood units)	7.4 ± 4.5		

Data are expressed as mean ± standard deviation or n (%). BMI, body mass index; SpO_2_, oxygen saturation; BNP, b-type natriuretic peptide; TRPG, tricuspid regurgitation peak gradient; TAPSE, tricuspid annular plane systolic excursion; FEV_1_, forced expiratory volume in 1 s; FVC, forced vital capacity; DLCO, diffusing capacity of lung for carbon monoxide; SPAP, systolic pulmonary artery pressure; DPAP, diastolic pulmonary artery pressure; mPAP, mean pulmonary artery pressure; PAWP, pulmonary artery wedge pressure; RAP, right atrial pressure; CO, cardiac output; CI, cardiac index; PVR, pulmonary vascular resistance.

**Table 2 life-12-00684-t002:** LPBV and ratio of LPBV in each lung of PH group and control group.

	PH Group N = 25	Control Group N = 20	*p* Value
LPBV			
Total PBV	38 ± 9	45 ± 8	0.024
Right PBV	38 ± 10	44 ± 8	0.039
Left PBV	39 ± 9	45 ± 9	0.021
Left-upper PBV	34 ± 10	47 ± 10	<0.001
Left-middle PBV	40 ± 9	45 ± 9	0.081
Left-lower PBV	43 ± 11	44 ± 10	0.701
Right-upper PBV	37 ± 10	47 ± 8	<0.001
Right-middle PBV	39 ± 10	43 ± 8	0.086
Right-lower PBV	40 ± 10	42 ± 9	0.516
The ratio of LPBV in each lung			
Right lung	1.00 ± 0.03	0.99 ± 0.02	0.228
Left lung	1.01 ± 0.04	1.02 ± 0.03	0.494
Right upper	0.95 ± 0.07	1.05 ± 0.10	<0.001
Right middle	1.01 ± 0.05	0.97 ± 0.04	0.012
Right lower	1.04 ± 0.07	0.93 ± 0.06	<0.001
Left upper	0.89 ± 0.14	1.05 ± 0.08	<0.001
Left middle	1.05 ± 0.08	1.01 ± 0.04	0.021
Left lower	1.12 ± 0.13	0.99 ± 0.07	<0.001

Data are expressed as mean ± standard deviation. PH, pulmonary hypertension; LPBV, lung perfused blood volume.

## Data Availability

Not applicable.

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
