# Peer review of "Quantification of Lung Perfusion Blood Volume in Dual-Energy Computed Tomography in Patients with Pulmonary Hypertension"

_life, 2022, doi:10.3390/life12050684_

Round 1

Reviewer 1 Report

It is my pleasure to review the manuscript entitled “Quantification of Lung Perfusion Blood Volume in Dual-Energy Computed Tomography in Patients With Pulmonary Hypertension”.

By Satoko Ugata, Satoshi Akagi et al.

 This study retrospectively verified the differences of lung perfusion blood volume (LPBV) obtained by dual-energy CT between the patients with PH and control without PH. The authors demonstrated lower total LPVB and lower LPVB of the upper lobe than those of control. In addition, a significant correlation between LPBV and lung perfusion scintigraphy was presented.

This study is of potential interest as the authors demonstrated the usefulness of LPBV for distinguishing PH from control. However, several minor concerns as below need to be addressed and included in the manuscript by authors to improve the quality of the articles.

  1. The PBV of upper lobe might be affected by artifact of the presence of iodine contrast agent in the superior vena cava and the subclavian vein. In this study, how the authors reduced the influences.

  1. Diffusing capacity of the lung carbon monoxide was extremely low for typical PAH. The authors should check the data again. If it is true, the reasons should be clarified. In pulmonary phenotype of idiopathic PAH, PBV must be influenced by impaired lung.

  1. In method section, the authors described that each lung was automatically divided into three lungs to calculate LPVB. However, each portion such as upper lung and lower lung was expressed as upper “lobe” or lower “lobe” in the manuscript. This reviewer recommends “lobe” should not be used because it is confusing and inaccuracy.

  1. In figure 6, this reviewer failed to confirm the gray line expressing 95% CI in the PDF file for review.

  1. How to calculate LPBV ratio should be explained in the method.

  1. The correlation between LPVB and lung perfusion scintigraphy was verified in this study, therefore, time difference in both tests should be clarified. The difference might have a bias against the results.

I hope these comments will be helpful. Thank you.

Author Response

Reviewer 1

We are grateful to reviewer for the critical comment and useful suggestion that have helped us to improve our manuscript. As indicated in the response that follow, we have taken all the comment and suggestion into account in the revised version of our manuscript.

Comment #1

The PBV of upper lobe might be affected by artifact of the presence of iodine contrast agent in the superior vena cava and the subclavian vein. In this study, how the authors reduced the influences

Response

  We agree with your opinion. The presence of iodine contrast agent in the superior vena cava and subclavian vein could cause beam-hardening artifacts and affect PBV of upper lung (J Comput Assist Tomogr, 41: 505-510, 2017 and Radiographics, 30: 3, 685-98, 2010). Although we used right median cubital to prevent the arrival delay of contrast medium and to reduce artifact in superior vena cava and the subclavian vein, it is limitation to measure PBV of upper lung. We described this limitation in discussion section (line 297-301).

Comment #2

Diffusing capacity of the lung carbon monoxide was extremely low for typical PAH. The authors should check the data again. If it is true, the reasons should be clarified. In pulmonary phenotype of idiopathic PAH, PBV must be influenced by impaired lung.

Response

  Thank you for checking DLCO. We are sorry that unit of DLCO in original manuscript was mL/min/mmHg, So the value was not %DLCO. %DLCO of the study was 54±18, which is almost within %DLCO range of typical PAH (J. Am. Card. 41 (2003) 1028-1035 and Respiratory Medicine 109 (2015) 1244-1249). We changed %DLCO in Table 1 of revised manuscript.

  We agree that PBV is influenced by impaired lung. We mentioned this point in limitation (line 297).

Comment #3

In method section, the authors described that each lung was automatically divided into three lungs to calculate LPVB. However, each portion such as upper lung and lower lung was expressed as upper “lobe” or lower “lobe” in the manuscript. This reviewer recommends “lobe” should not be used because it is confusing and inaccuracy.

Response

  We agree that the word “lobe” make readers confuse. We changed the “lobe” to “lung” in revised manuscript.

Comment #4

In figure 6, this reviewer failed to confirm the gray line expressing 95% CI in the PDF file for review.

Response

  We are sorry about figure 6. We changed new version of figure 6 with expressing 95% CI.

Comment #5

How to calculate LPBV ratio should be explained in the method.

Response

  We are sorry not to explain the definition of LPBV ratio. We defined LPBV ratio as the value of LPBV divided by total LPBV. We explained how to calculate LPBV ratio in method section (line 96-97).

Comment #6

The correlation between LPVB and lung perfusion scintigraphy was verified in this study, therefore, time difference in both tests should be clarified. The difference might have a bias against the results.

Response

  We agree with our opinion. We checked the time difference between DECT and lung perfusion scintigraphy. Fifteen patients underwent DECT within one week of lung perfusion scintigraphy. Three patients underwent DECT within one month of lung perfusion scintigraphy. Two patients underwent DECT after one year of lung perfusion scintigraphy. We added these time difference in results section (line 210-212). These time differences would have a bias against the results. We mentioned this point in limitation (line 302-304).

Reviewer 2 Report

I have read with interest this paper. This retrospective study analyses 25 patients affected by pulmonary hypertension (PH) versus 20 controls, in order to investigate the usefulness of Dual-energy computed tomography (DECT) in detecting lung perfused blood volume (LPBV). Moreover, results obtained with DECT have been compared to lung scintigraphy.  The total LPBV was significantly lower in the PH group than control group

I have really appreciated this work: it is a well written paper on a well conducted SPECTAlung program.

  • Introduction: offers a concise overview on the issue.
  • Materials and Methods: Well described and accurate, just a comment:
    • Time period: 2013-2014. As the long length of time, why the authors have decided to not implement the sample size?
  • Results: accurate and detailed presentation.
  • Discussion: clear and comprehensive
  • Tables and figure: interesting, appealing and not redundant

In my opinion, a brief paragraph on scientific soundness and possible improvement given by this imaging technique (e.g. in diagnosis and therapeutic approach) should be added.

Author Response

Reviewer 2

We are grateful to reviewer for the critical comment and useful suggestion that have helped us to improve our manuscript. As indicated in the response that follow, we have taken all the comment and suggestion into account in the revised version of our manuscript.

Comment #1

In method section, why the authors have decided to not implement the sample size?

Response

  We agree with the reviewer’s comment on importance of study sample size. Because this study was a retrospective observational study, however, we could not collect much more subjects than that which we could access as the source population of this study. Therefore, we added the description related to this issue in conclusion section as follows: “Further investigation with larger sample size is warranted.” (line 317).

Comment #2

A brief paragraph on scientific soundness and possible improvement given by this imaging technique (e.g. in diagnosis and therapeutic approach) should be added

Response

  Thank you for your suggestion. We added followed sentences in discussion section (line 305-311). “LPBV obtained from DECT is an excellent diagnostic imaging technique that could contribute to the diagnosis of PH because it could visualize and quantify regional perfusion decline in addition to anatomical information obtained from good spatial resolution. Especially in situations where pulmonary blood flow scintigraphy could not be performed, it might be an alternative imaging examination which could evaluate regional perfusion. Although it is a small study, we hope present study would be useful in clinical practice.”

Round 2

Reviewer 2 Report

The authors have clarified and responded to my comment. No others suggestion/concern from my side. Well done.